# Vitamin D Implications and Effect of Supplementation in Endocrine Disorders: Autoimmune Thyroid Disorders (Hashimoto’s Disease and Grave’s Disease), Diabetes Mellitus and Obesity

**DOI:** 10.3390/medicina58020194

**Published:** 2022-01-27

**Authors:** Dorina Galușca, Mihaela Simona Popoviciu, Emilia Elena Babeș, Mădălina Vidican, Andreea Atena Zaha, Vlad Victor Babeș, Alexandru Daniel Jurca, Dana Carmen Zaha, Florian Bodog

**Affiliations:** 1Department of Medical Disciplines, Faculty of Medicine and Pharmacy, University of Oradea, 410073 Oradea, Romania; dorinagalusca@gmail.com (D.G.); elapopoviciu@yahoo.com (M.S.P.); eebabes@uoradea.ro (E.E.B.); babesvlad@gmail.com (V.V.B.); 2Department of Preclinical Disciplines, Faculty of Medicine and Pharmacy, University of Oradea, 410073 Oradea, Romania; mada_vidican@yahoo.ro (M.V.); alexjurca@yahoo.co.uk (A.D.J.); 3Faculty of Medicine, “Iuliu Hațieganu” University of Medicine and Pharmacy, 400000 Cluj Napoca, Romania; andreeaatenazaha@gmail.com; 4Department of Surgery, Faculty of Medicine and Pharmacy, University of Oradea, 410073 Oradea, Romania; fbodog@gmail.com

**Keywords:** vitamin D, Hashimoto’s disease, Grave’s disease, diabetes mellitus

## Abstract

*Background and Objectives*: Vitamin D deficiency is involved in numerous pathologies, including endocrine pathology. The purpose of this review consists of presenting the role of vitamin D in the pathophysiology of certain endocrine disorders, autoimmune thyroid disorders (Hashimoto’s disease and Grave’s disease), diabetes mellitus, and obesity, and whether its supplementation can influence the outcome of these diseases. *Materials and Methods*: Review articles and original articles from the literature were consulted that corresponded with the thematic. *Results*: Vitamin D deficiency is frequently encountered in endocrine disorders and supplementation restores the normal values. In Hashimoto’s disease, vitamin D deficiency appears to be correlated with a higher titer of anti-TPO antibodies and with thyroid volume, and supplementation was associated with reduction of antibodies in some studies. In other studies, supplementation appeared to reduce TSH levels. In Grave’s disease, there was a significant correlation regarding vitamin D levels and thyroid volume respective to the degree of exophthalmos. In diabetes mellitus type 2 patients, supplementation led to some improvement of the HOMA-IR index and HbA1c, whereas obesity data from literature do not report significant beneficial findings. *Conclusions*: Vitamin D deficiency is highly prevalent in endocrine disorders and its supplementation appears to have numerous beneficial effects.

## 1. Introduction

Recently, meta-analyses and observational studies revealed that the prognosis of various common diseases, including endocrine, autoimmune, and chronic disorders and even cancer progression, was related to vitamin D plasmatic concentration. It was proven that immune system cells (B cells, T cells, and antigen-presenting cells) are particularly capable of producing the active metabolite of calciferol (a substance presenting immunomodulatory proprieties), which is possible because of 1α-hydroxylase (CYP27B1) gene expression. Furthermore, the expression of the vitamin D receptor (VDR) on the cell surface hints at this molecule actioning locally in the immune response, findings which are supported by proven relationships between VDR or CYP27B1 gene polymorphisms and several autoimmune diseases’ pathogenesis [1]. Human experimental studies demonstrated that vitamin D supplementation was correlated with beneficial effects such as reducing disease severity in autoimmune disorders; however, its optimal serum concentrations are currently under debate [1].

Vitamin D has essential roles in calcium regulation, phosphorus homeostasis, and promoting bone health. Evidence from animal and human studies prove that vitamin D is involved in the pathogenesis of various endocrine disorders. High interest in vitamin D among researchers and clinicians relies on frequent publications that illustrate the pleiotropic effects of the compound and higher prevalence of hypovitaminosis among healthy individuals. These endocrine conditions include primary hyperparathyroidism, type 1 diabetes (T1DM), type 2 diabetes (T2DM), autoimmune thyroiditis, adrenal gland diseases, and polycystic ovary syndrome (PCOS) [2].

One of the main findings is that vitamin D inhibits the synthesis of some inflammatory cytokines such as interleukin (IL)-1, IL-6, IL-8, IL-12, and tumor necrosis factor (TNF)-α. Due to the decreased expression of major histocompatibility complex (MHC) class II proteins, co-stimulatory elements, and IL-12, adequate levels of vitamin D suppress dendritic cell differentiation and maturation of these cells [3]. Another aspect consists of influencing the activity of the regulatory T cells by vitamin D, leading to a decreased T-cell-dependent immune effect in autoimmune diseases. In particular, T cells and B cells respond to thyroid antigens in subjects who are genetically susceptible, and this can trigger the onset of hyperthyroidism. Serum levels of 1,25(OH)_2_-Vitamin D_3_ < 20 ng/mL were correlated with increased synthesis of thyroid autoantibodies such as anti-thyroid peroxidase (TPOAb) and anti-thyroglobulin (TgAb) [3].

Vitamin D has an important impact on metabolic disorders. There is various evidence regarding the improvement of the metabolic profile of subjects receiving vitamin D supplementation, especially by decreasing lipids—total cholesterol (TC), low-density lipoprotein cholesterol (LDL-cholesterol), triglycerides (TG), and other parameters like glycosylated hemoglobin (HbA1c), along with a significant decrease in the insulin resistance indicator—HOMA-IR—in T2DM patients. The mechanism through which metabolic risk reduction occurs is not fully elucidated. VDR and vitamin D-metabolizing enzymes have been detected in different types of cells, including pancreatic β cells and insulin-responsive effectors like adipocytes. Vitamin D is stored adipocytes, which are cells that produce a significant source of adipokines and cytokines, which contribute to systemic inflammation. The fact that obesity, especially visceral, is one of the major risk factors for T2DM is well known [4]. Similar to obesity, patients with non-alcoholic fatty liver disease (NAFLD) tend to have vitamin D deficiency, and this is associated with the risk of various infections [5].

The purpose of this review is to present the status of vitamin D, the effects of its deficiency, and the beneficial role of its supplementation in certain endocrinological disorders that are very commonly encountered in clinical practice.

## 2. Material and Methods

We performed research on the relevant literature and selected scientific publications addressing the correlation between the above-mentioned endocrine pathology and vitamin D. The PubMed and Web of Science databases were searched for relevant information related to this topic. The method used to search for relevant information was done with the help of the “AND” operator and specific keywords (“autoimmune thyroid disorders” AND “vitamin D”, “Hashimoto’s diseases” AND “vitamin D”, “Graves’ disease” AND “vitamin D”, “type 2 diabetes mellitus” AND “vitamin D”, “obesity” AND “vitamin D”, etc.) (Table 1). The articles considered eligible were initially selected based on both their title and abstract; then, a complex analysis of their content was performed, and the most relevant and informative data and results were extracted. Our aim was to discuss autoimmune thyroid pathology, diabetes mellitus type 2, and obesity because we work in tertiary care hospitals in Oradea, Romania, where we encounter the above-mentioned endocrinological diseases most often in the diabetes clinic, internal medicine clinic, and endocrinology clinic. The diseases were chosen according to the observed frequency of their apparition in the hospitals that we work in and our continuous preoccupation with the possibility of improving the outcome of these patients by assessing and, where necessary, correcting vitamin D status.

## 3. Vitamin D General Implications in Health and Daily Requirements

Vitamin D affects the transcription of some genes through genomic and non-genomic mechanisms. The action is mediated by VDR, which is a type of nuclear receptor that acts as a ligand-activated transcription factor. After generating the active form of vitamin D, 1,25(OH)_2_-vitamin D_3__,_ binds to VDR, resulting in a heterodimeric molecule through interaction with the retinoid receptor (RXR). This complex molecule is further translocated to the nucleus. There, the binding of this complex to the vitamin D-responsive elements (VDRE) in the promoter region of the vitamin D-responsive genes occurs. Chromatin remodeling is facilitated by co-regulatory elements that produce changes of epigenetic histones as well as local RNA polymerase II recruitment; these effects are based on the connection existing between the 1,25(OH)_2_-vitamin D_3_–VDR–RXR and VDRE. All these processes influence the expression of some genes, including those incriminated in cell proliferation and differentiation, immunomodulatory response, and angiogenesis [6,7].

Activation of a significant number of signaling molecules (i.e., phosphatidylinositol-3 kinase, phospholipase C (PLC), Ca^2+^-calmodulin kinase II (CaMPKII), mitogen-activated protein kinases (MAPKs), src, protein kinases A and C) are responsible for the non-genomic actions of vitamin D. All these kinases target transcription factors (i.e., SP1, SP3, and RXR), which ultimately engage VDRE in the activation of vitamin D-responsive genes. The synthesis of second messengers (such as cyclic AMP, Ca^2+^, fatty acids, and 3-phosphoinositide) is also promoted in the aforementioned process. Different types of cells and development stages determine the range of signaling molecules [8].

Vitamin D levels can be monitored by detecting the concentration of its circulating metabolite 25(OH)-vitamin D. This substance illustrates the vitamin D intake through diet and cutaneous production, and since it has a long half-life (10–19 days) it can be reliably used as a diagnostic indicator in several guidelines, such as the ones developed by the American Society of Endocrinology and the Institute of Medicine [9]. Serum concentrations of 25(OH)-vitamin D and 1,25(OH)_2_ -vitamin D_3_ can be measured, and the first one represents a reliable evidence of vitamin D metabolic status. The latter offers no valuable clues and may often be within a normal or elevated range as a result of the secondary hyperparathyroidism that appears in vitamin D deficiency [10]. Previous guidelines define vitamin D deficiency as 25(OH)–vitamin D serum concentration less than 50 nmol/L (20 ng/mL). In insufficiency, these values can go as high as 50–75 nmol/L (20–30 ng/mL) [11]. However, nowadays values such as 50–125 nmol/L (20–50 ng/mL) are generally considered safe with regards to musculoskeletal health. Deficiency and insufficiency criteria have dropped to values below 30 nmol/L (12 ng/mL) and 30–50 nmol/L (12–20 ng/mL), respectively [12]. Prevention of vitamin D deficiency is usually achieved by daily doses of 600–800 UI, recommended by several nutrition societies in Europe and in the USA [13]. A quantity of 4000 IU/day is considered the upper limit regarding the safety of cholecalciferol [14].

## 4. Role of Vitamin D in Thyroid Autoimmune Diseases and Effects of Supplementation

### 4.1. Implications in Hashimoto’s Disease

Even though there is increasing evidence that hyperthyroidism results from the interaction between genetic susceptibility and environmental risk factors, the exact mechanism remains unknown [15].

This disease has been linked to the occurrence of papillary thyroid carcinoma by several clinical studies, and neuroinflammation that caused emotional alterations was also reported in one study despite normal thyroid function. Its incidence has been increasing recently and is unfortunately largely asymptomatic. Only in later stages can patients present with abnormal thyroid function, other associated conditions, and even malignant tumors, therefore causing psychological distress and economic burden. For this reason, understanding the appearance of hyperthyroidism in healthy individuals and its relationship with appropriate indicators is essential for general public health [16,17]. Hashimoto’s disease (HD) is an autoimmune thyroid disease that presents histologically as a chronic lymphocytic thyroiditis. Its etiology remains unknown; however, recent studies have shown that the most important pathogenic factors are Th1/Th2 imbalance as well as increased activity of Th1 cells. The immune response in HD may be inhibited by vitamin D acting as an immunosuppressive agent [18].

Biological effects of active vitamin D are consequences of some important signaling molecules activated by the interaction between vitamin D and receptors. 1,25(OH)_2_-vitamin D_3_ binds to the VDR on target cells, forming a heterodimeric molecule, and this ensures protein-dependent transport [19].

Vitamin D has been proven to be an essential element in macrophage maturation. In addition, it was also noted to promote monocyte differentiation in macrophages, increase phagocytosis and chemotaxis, and enhance mononuclear macrophages’ anti-tumoral effects (Figure 1)Vitamin D can inhibit TLR2 and TLR4 (Toll-like receptors) on monocyte surface expression, which implies inhibiting the identification of pathogen-associated molecular patterns by these receptors, consequently reducing immune responsiveness and the production of inflammatory cytokines and also preventing an exaggerated immune response. The overall result is reduced inflammation [20].

The effects of 1,25(OH)_2_-vitamin D_3_ on dendritic cells have been illustrated by various studies. The first step consists of the inhibition of the p38 mitogen-activated protein kinase (MAPK) and nuclear factor kB (NF-kB) signal pathways by vitamin D, an action that influences the synthesis of interleukins in dendritic cells. Secondly, it increases anti-inflammatory cytokine production (such as IL-10) and T-cell inhibitory molecules (programmed death-1). Moreover, differentiation of T-helper lymphocytes is impaired due to the decreased synthesis of pro-inflammatory cytokines like IL-12, IL-23, TNF-α, and IFN-γ [21].

One study enrolled 75,436 patients who underwent physical examination and serum hormone level testing; however, only 5656 had their 25(OH)-vitamin D levels measured and even fewer ended up being enrolled-5230. These levels were higher in patients not presenting with hyperthyroidism. Hyperthyroidism was significantly correlated by means of multiple regression analysis with gender (male), body mass index, abdominal circumference, and TSH circulating levels. The latter had higher levels in the insufficiency and deficiency 25(OH)-vitamin D group. Inversely, free triiodothyronine and thyroxine levels were lower in these patients than those with sufficiency. There was an increase in 25(OH)–vitamin D levels by 1 ng/mL from baseline, an increase in FT4 levels of 2.78 ng/dL, and a decrease of 0.17 mIU/L in TSH levels. Reduced 25(OH)-vitamin D levels were present in subjects diagnosed with hyperthyroidism; the former was positively correlated with FT3 and FT4 levels. TSH represents an independent predictor for hyperthyroidism occurrence [22].

In another study, weekly supplementation of 50,000 IU were given to deficient patients for eight weeks, at the end of which all measurements were repeated. HD patients had statistically significant lower levels of vitamin D compared to the control group (9.37 ± 0.69 ng/mL vs. 11.95 ± 1.01 ng/mL, *p* < 0.05) and the euthyroid subgroup had significantly lower thyroid antibody titers after supplementation. The same subgroup had improved HDL cholesterol levels. Mean free thyroxine levels were 0.89 ± 0.02 ng/dL in the HD group and 1.07 ± 0.03 ng/dL in the control group; the *p*-value was <0.001, which suggests significant differences between HD and healthy patients. Significant differences were also found between the former’s and latter’s mean thyroid volumes (7.71 ± 0.44 mL vs. 5.46 ± 0.63 mL, respectively; *p* < 0.01). An important conclusion was that vitamin D deficiency represents a common finding in HD patients; therefore, treating it may be beneficial in the progression of hypothyroidism and also in cardiovascular risk reduction. Effective management of these cases may be critical [23].

Vitamin D’s active form is a steroid hormone involved in bone and mineral balance, and recently, the anti-inflammatory and immunomodulation properties were discovered. Most of the vitamin’s effects are mediated by its receptor (VDR), and its main regulators are CYP27B1 hydroxylase and the vitamin D-binding protein [24,25]. It has been found that allele variations in the VDR gene play a role in susceptibility to a number of autoimmune endocrine disorders [26]. A link between autoimmune thyroid disorder AITDs and VDR gene polymorphisms was recently proven by several genetic studies. An important finding consists of the interference of proteins and several enzymes with vitamin D functions. Horst-Sikorska et al. demonstrated a relationship between VDR polymorphism and susceptibility to thyroid autoimmunity by linking F allele carriers to VDR–Fokl polymorphism and Graves’ disease [27].

Moreover, the vitamin D-binding protein illustrated genetic polymorphism in Polish and Japanese subjects with AITDs, whereas the CYP27B1 hydroxylase polymorphism was identified as a predisposing factor for Hashimoto’s thyroiditis and Graves’ disorder in German patients [28,29,30].

In the study conducted by Kivity et al., 50 patients with autoimmune thyroid diseases, 42 with non-autoimmune thyroid diseases, and 98 healthy subjects were enrolled and had their serum 25(OH)-vitamin D levels measured through LIAISON chemiluminescence immunoassay. Vitamin D deficiency was reported as levels lower than 10 ng/mL. Other measurements included antithyroid antibodies, biomarkers evaluating thyroid functions, and demographic data. Vitamin D deficiency had significantly higher rates in AITD patients compared to the control group (72% vs. 30.6%; *p* < 0.001), and the same was true upon comparing HD patients to non-AITDs (79% vs. 52%; *p* < 0.05). Significantly low vitamin D levels have also been linked to the detection of antithyroid antibodies (*p* = 0.01) and abnormal markers of thyroid function (*p* = 0.059), thus suggesting its involvement in AITD pathogenesis and highlighting the necessity of supplementation [31].

Mazokopakis et al. showed an inverse relationship regarding serum 25(OH)-vitamin D levels and anti-TPO antibody production in 218 patients with hyperthyroidism and normal thyroid function. Vitamin D-deficient patients had significantly higher anti-TPO antibody levels than patients without this deficiency. A significant decrease (20.3%) in serum anti-TPO levels was noted in 186 patients who received oral vitamin D_3_ supplementation (1200–4000 IU/day) for 4 months [32].

Chaudhary et al. studied 100 patients who had recently been diagnosed with autoimmune thyroid disease and discovered that those in the lowest 25(OH)–vitamin D quartile had the highest levels of anti-TPO antibodies (*p* = 0.084). A three-month follow-up showed significant decreases in the levels in patients who were administered vitamin D_3_ supplementation for 8 weeks (60,000 IU/week). Comparing this group to the patients without vitamin D supplementation, there was a 46.73% decrease versus 16.6%, with this difference being statistically relevant (*p* = 0.028). The former also reported a higher number of responders (≥25% decrease in anti-TPO levels) than the latter (68% vs. 44%; *p* = 0.015) [33].

A randomized clinical trial study done by Chahordoli et al. included 42 women with HD, who were divided into vitamin D and placebo groups. The first group was assessed to receive 50,000 IU of vitamin D, whereas the second one received placebo pearls each week for 3 months. Laboratory parameters such as 25(OH)–vitamin D, calcium, anti-TPO, anti-Tg, thyroid hormones, and TSH were assessed at the beginning and end of the study through enzyme-linked immunosorbent assays. Significant reductions in anti-Tg and TSH were found in the first group after treatment, but there were no relevant differences in anti-TPO levels between the two groups (*p* = 0.08). Similarly, thyroid hormone levels changed insignificantly. It was thus concluded that vitamin D supplementation was useful in decreasing disease activity; nonetheless, additional evidence promoted by well-controlled longitudinal trials is essential to establish its relevance in clinical practice [34].

The study designed by Behera et al. included patients with hyperthyroidism, 23 of whom received doses of 60,000 IU per week of vitamin D for 8 weeks and then the same dose once per month for 4 months. TPO antibody and thyroid hormone serum concentrations were measured again after 6 months. Increases in vitamin D levels were statistically significant: from 15.33 ± 5.71 ng/mL to 41.22 ± 12.24 ng/mL. TPO antibody titers illustrated a statistically significant increase from 746.8 ± 332.2 to 954.1 ± 459.8 IU/mL (*p* = 0.006), whereas TSH values demonstrated a statistically significant decrease from 7.23 ± 3.16 to 3.04 ± 2.62 mIU/L (*p* = 0.01) [35].

In a meta-analysis that included 25 studies comprising 2695 cases and 2263 controls, it was discovered that HD patients had lower 25(OH)–vitamin D serum levels in comparison to the control groups; however, there was significant heterogeneity between studies (Cohen’s D = 0.62; 95% CI 0.89–0.34; *p* = 1.5 × 10^−5^). In vitamin D-deficient individuals compared to the control groups, an odds ratio of 3.21 of having HD (1.94–5.3; *p* = 5.7 × 10^−6^) was determined. These findings were consistent across all included studies—European and Asian, adult and pediatric, and moderate- and high-quality studies. Higher differences in 25(OH)D values between the groups were related to near-equatorial latitudes (<35° N/S, *p* = 3.4 × 10^−4^) and moderate-income economies (gross national income 1000 < USD < 12,000, *p* = 0.012). In univariate meta-regression, Cohen’s d was found to be lower in correlation with higher latitude (*p* = 0.0047) or higher mean body mass index (*p* = 0.006 in 10 studies). Gross national income (*p* = 3.5 × 10^−6^) and mean serum thyrotropin in the affected subjects (*p* = 0.017 in 21 studies) revealed nonlinear moderation. The main finding illustrates a significant connection between HD and 25(OH)–vitamin D and partly explains previous mixed evidence, providing a major contribution in revealing the factors involved in heterogeneity, highlighting under which conditions there is the strongest link [36].

In a meta-analysis done by Taheriniya et al., even though 6123 datasets were reviewed, the inclusion criteria for this systematic review and meta-analysis were met in 42. Lower vitamin D levels were correlated with autoimmune thyroid diseases (*p* = 0.013), Hashimoto’s thyroiditis (*p* < 0.001), and hypothyroidism (*p* = 0.03). The same could not be said about patients with Graves’ disease (*p* = 0.06), although this relationship was only significant in patients older than 40 years [37]. These data are summarized in the Figure 2.

### 4.2. Implications of Vitamin D Deficiency in Graves’ Disease

Graves’ disease (GD) is another relatively frequent autoimmune thyroid disorder associated with the production of antibodies against the TSH receptor (TRAb), causing hyperthyroidism. Multiple genetic and environmental factors contribute to its pathogenesis. Recently, vitamin D has been shown to play a part, as serum concentrations tend to be decreased and related to thyroid volume in female subjects with new-onset Graves’ disease [38].

Data from the literature identified a number of polymorphisms related to the vitamin D gene, including VDR and vitamin D-binding protein, and these polymorphisms have been linked to GD [28].

In BALB/c mice, it has been illustrated that deficiency of vitamin D modulates hyperthyroidism in GD through immunization of the thyrotropin receptor. It also inhibits CXCL10 in the human thyroid cells, a Th1 chemokine that has been recognized to play a crucial part in GD pathogenesis. Moreover, vitamin D inhibits the inflammatory responses involved in the etiology of GD and metabolic syndrome [39,40].

The serum concentration of vitamin D showed a significantly lower value in GD subjects who were not in remission compared to those who were. To the best of our knowledge, this was the first study to demonstrate differences related to remission status in women. The durations of hyperthyroidism from GD onset were significantly longer in the group with no remission that the other. However, differences in onset were not calculated [41].

Despite its main finding, it is also important to note that this cross-sectional survey included a limited number of participants and therefore had limited value in establishing whether vitamin D is directly involved in pathogenesis and prognosis of GD [41].

This randomized prospective study done by Sheriba et al., which included 60 adults with GD (aged 20–40), divided them into two groups: In the first one, 20 patients received 30 mg methimazole daily, and in the second one, 40 patients received the same dose of methimazole and 200,000 IU monthly of vitamin D3 for 3 months. Follow-up was conducted over a 3-month period. All participants presented with a form of hypovitaminosis—73.9% males and 54.1% females with deficiency and 26.1% males and 45.9% females with insufficiency. A strong relationship was revealed between vitamin D levels, thyroid volume, and the degree of exophthalmos. Lowering thyroid volume was revealed upon supplementation in the second group and positive effects on the exophthalmos degree were also discovered [41].

Recent-onset GD was analyzed in 292 patients and 2305 controls by means of vitamin D levels by Planck et.al. Connections between GD/Graves’ ophthalmopathy and VDR single nucleotide polymorphisms between vitamin D-binding protein and *CYP27B1*, respectively, were examined in 708 patients and 1178 controls. The results suggested significantly lower vitamin D values (*p* < 0.001) in the GD group (55.0 ± 23.2 nmol/L) compared to healthy patients (87.2 ± 27.6 nmol/L). Values of thyroid hormones such as free thyroxine or free triiodothyronine, thyrotropin receptor antibodies, relapse upon antithyroid medication discontinuation, and Graves’ ophthalmopathy did not reveal an association with vitamin D levels at diagnosis. Two of the VDR’s SNPs were linked to GD: rs10735810 (OR = 1.36, 95% CI: 1.02–1.36, *p* = 0.02) and rs1544410 (OR = 1.47, 95% CI: 1.03–1.47, *p* = 0.02). Neither the first nor the second SNP had differences in mean vitamin D concentrations across genotypes [42].

Meta-regression and sensitivity analysis were also performed by combining the effect sizes from 26 studies by Xu et al. A pooled connection of standard mean difference (SMD) = −0.77 (95% CI: −1.12, −0.42; *p* < 0.001) was found, and random effect analysis favored the low vitamin D concentrations. Meta-regression concluded that heterogeneity was mainly influenced by the assay technique (*p* = 0.048). GD patients were more predisposed to vitamin D deficiency in comparison with the control group (OR = 2.24, 95% CI 1.31–3.81); this finding revealed a high heterogeneity. Thus, low vitamin D status was further proven to increase GD risk [43].

The study designed by Yasuda et al. illustrated high statistical prevalence (*p* < 0.05) of vitamin D deficiency in GD (65.4%) compared to the control group (32.4%). Significant correlation was found regarding vitamin D levels, calcium levels (r = 0.49; *p* < 0.05), and unchanged parathyroid hormone concentrations (r = −0.50, *p* < 0.05). Furthermore, significant correlations between thyroid volume and 25(OH)–vitamin D levels were discovered (r = −0.45; *p* < 0.05) but the same could not be established for TRAb values and thyroid function [44]. Conclusions of these studies can be seen in the Figure 3.

## 5. Vitamin D Deficit in Type 2 Diabetes Mellitus and Effects of Supplementation

Insulin secretion is largely influenced by 1,25(OH)_2_-cholecalciferol due to its calcium-increasing influx effect on pancreatic β cells. This ion influx is the last process in the insulin secretion cascade as a response to high blood sugar levels. The final result consists of insulin exocytosis from its vesicles [45].

In tissues, the regulation of insulin action is accompanied by vitamin D directly by the effects of insulin receptors or indirectly through the stimulation of peroxisome proliferator-activated receptors (PPARs). These act like nuclear transcription factors and play a major part in controlling fatty acid metabolism in both adipocytes and myocytes [46].

Insulin resistance is believed to be mainly caused by increasing severity of chronic inflammation, and it seems that mitochondria impairment plays a pivotal role in the mechanisms leading up to this. Vitamin D decreases the level of pro-inflammatory cytokine production, decreasing properties and facilitating anti-inflammatory cytokine synthesis. Vitamin D deficiencies are usually linked to pro-inflammatory states [47,48].

Another substantial role of vitamin D is in epigenome sustenance. The epigenetic changes that accompany DM2 and insulin resistance are often linked to DNA hypermethylation, which causes multiple gene inactivation. This can be counteracted by vitamin D through increased DNA demethylase expression [49].

Insulin sensitivity in peripheral cells may be enhanced by vitamin D by several mechanisms. It is apparent that 1,25(OH)_2_-vitamin D_3_ is associated with increased insulin sensitivity by stimulating insulin receptor expression upon binding to a VDRE in the human insulin receptor gene promoter. This may also be a result of PPARδ activation [50].

Another indirect mechanism may involve calcium homeostasis regulation. It is a well-known fact that calcium modulates intracellular pathways in insulin-responsive tissues and has a very narrow range in which optimal function occurs. As a result, small alterations in vitamin D levels may be sufficient to impair insulin signal transduction, and this in turn decreases the activity of the glucose transporter [51]. 

Vitamin D acting as a mediator in ameliorating insulin sensitivity is connected to insulin signaling. When transcriptional activation of the IR gene occurs thanks to 1,25(OH)_2_-vitamin D_3_, there is an increased number of IRs on the surface of insulin-responsive cells. Therefore, proper insulin signaling is ensured by IR gene upregulation, and thus insulin sensitivity is maintained by calcitriol [52].

Vitamin D deficiency was found to be incriminated in insulin resistance onset because of the reduced expression of IR even though the activation of IR expression in the hepatic cells mediated by vitamin D has been reported to offer inconclusive results. George et al. illustrated the fact that liver expression of IR was upregulated by vitamin D supplementation in streptozocin-induced diabetic mice. Another aspect consists of the inability to reveal changes in IR expression in mice that were fed high- or low-fat diets or in streptozocin-induced diabetic rats after supplementation [53].

A systematic review comprising 23 RCTs included a total of 1797 T2DM patients. The mean changes in 25(OH)D levels revealed a variation in the values ranging from 1.8 ± 10.2 nmol/L till 80.1 ± 54.0 nmol/L. HbA1c was included in 19 studies as an outcome variable; however, when combined, these studies demonstrated no significant effects in subjects with vitamin D supplementation compared to placebo. A significant effect (*p* = 0.003) was detected in baseline fasting glucose due to a subgroup of four studies with a mean baseline value of HbA1c ≥ 8% (64 mmol/mol); SMD = 0.36; 95% CI: 0.12–0.61 [54].

Twenty studies with 2703 participants were included in the meta-analysis. Vitamin D supplementation significantly improved serum 25(OH)-vitamin D levels (weighted mean difference (WMD) = 33.98; 95% CI 24.60–43.37) and HOMA-IR (SMD = −0.57; 95% CI −1.09~−0.04), but there were no other major influences on the outcome [55].

Nineteen RCT studies that gathered a total of 747 intervention subjects and 627 placebo controls were selected for this meta-analysis. The short-term vitamin D supplementation group revealed an improvement in HbA1c values, insulin resistance, and insulin in comparison with the values registered in the placebo group. The SMDs of HbA1c, insulin resistance, and insulin were −0.17 (−0.29, −0.05), −0.75 (−0.97, −0.53), and −0.57 (−0.78, −0.35), respectively, all with *p*-values below 0.05 (95% CI). However, long-term follow-up consequent to vitamin D supplementation was not associated with major positive effects. This short-term treatment had a positive impact regarding HbA1c values, insulin resistance, and insulin in T2DM patients, suggesting that vitamin D may represent an additional therapeutic target in diabetes [56].

Eight trials that included a total of 4896 participants were identified as eligible. A significant reduction in the risk characteristic for T2DM subjects (risk ratio (RR) 0.89) was discovered due to vitamin D supplementation (95% CI 0.80–0.99; *I*^2^ = 0%). The effect on reducing relative risk was stronger in nonobese individuals (RR 0.73 (95% CI 0.57–0.92)) without revealing the same activity in obese subjects (RR 0.95 (95% CI 0.84–1.08)) (P_interaction_ = 0.048). Reversal of prediabetes to normoglycemia was illustrated in 116 of 548 (21.2%) participants from the intervention group and in 75 of 532 (14.1%) subjects from the control group. Supplementation was associated with the reversal of prediabetes to normoglycemia (RR 1.48 (95% CI 1.14–1.92); *I*^2^ = 0%.) The study concluded that a major decrease in the risk involved in T2DM occurrence was seen in vitamin D supplementation, and the reversion rate of prediabetes to normoglycemia was high in prediabetic individuals. The impact of the prevention of T2DM may be lower in obese patients. Further meta-analysis examining individual participant data is necessary to reveal these insights [57].

## 6. Deficit in Obesity and Impact of Supplementation on Weight Loss

It has been postulated that vitamin D deficiency induces high levels of parathyroid hormone, which promotes lipogenesis through an increase in calcium inflow in adipocytes; however, these data are merely experimental. Another more reasonable explanation involves adipogenesis being inhibited by the active form of vitamin D through the VD receptors [58].

Blumberg et al. showed that VDRs inhibited the differentiation of 3T3-L1 preadipocytes in the presence of the active form of vitamin D through the downregulation of the adipocyte-stimulating transcription factor C/EBPβ [59].

Moreover, the WNT/β-catenin pathway can be maintained by 1,25(OH)_2_-vitamin D_3_, and since it is downregulated during adipogenesis, it may inhibit the process of adipogenesis [60].

In conclusion, lower VD levels may promote the differentiation of pre-adipocytes into adipocytes. Furthermore, two longitudinal studies proved that these concentrations predispose individuals to obesity and promote greater weight gain than in patients with higher baseline levels of VD [61,62].

Vitamin D levels showed a weak but statistically significant inverse correlation with body mass index in 34 cross-sectional studies [63].

The link was observed in both males and females in developed countries and strictly in males in developing countries.

A causal association between vitamin D levels and obesity was explored by a large meta-analysis consisting of 21 population-based studies with a total of 42,024 individuals [64].

Large cohorts were observed by a bidirectional mendelian analysis that showed that obesity could lead to low vitamin D levels and not the opposite. The increase with a unit of the BMI (1 kg/m^2^) was associated with a reduction of 1.15% in 25(OH)–vitamin D. This connection was evaluated after the adjustment of several factors such as age, gender of the subjects, and other confounders. The connection between obesity, defined as BMI > 30 kg/m^2^, and vitamin D deficiency and insufficiency across all age groups was revealed in a meta-analysis that included 12 observational studies (RR 1.52, 95% CI: 1.33–1.73) [65].

The deficiency of vitamin D was also reported to be more prevalent in overweight and obese patients (24% and 35%, respectively, more than in normal-weight subjects) by another meta-analysis consisting of 23 observational studies. Severe obesity reports were excluded from this analysis [66].

Low production of vitamin D been shown to be independently associated with obesity irrespective of latitude, age, defining criteria of deficiency, and the country’s development status. A recently published meta-analysis that included 55 studies confirmed the inverse link between BMI and vitamin D. This correlation was also shown to be stronger in T2DM patients and notably weaker in non-diabetic patients within a normal BMI range. In the first group, steep increases in association strength in patients with BMI > 30 kg/m^2^ were noted [66].

A considerable number of interventional clinical studies illustrated the effects of vitamin D supplementation on obesity, especially RCTs. In recent years there have been eight meta-analyses published on this subject, and one meta-analysis that included 12 RCTs researched the impact of supplementation on obesity through various measures such as BMI, PFM, and FM in the absence of caloric restriction [67]. Various doses (20–7000 IU daily, 20,000–40,000 IU weekly, 50,000 IU every 20 days, or 120,000 IU three times every two weeks) of calcitriol were administered to the intervention group. The subjects were mainly obese or overweight adults, but two studies also included children older than 10. Studies took place for one month to three years. Meta-analysis suggested that none of the adiposity measures were decreased by supplementation. Moreover, body weight was not influenced by changes from the baseline or by normal vitamin D status achievement. Interestingly however, eight out of 12 RCTs showed reductions in BMI, but they were not statistically significant. Potential confounding factors were age and gender as younger individuals, and females were more likely to undergo reductions in FM [67].

In another meta-analysis that included 26 RCTs (and also eight out of 12 of the ones analyzed by Pathak et al.), 42,430 adults and 12 months of treatment did not show significant improvements regarding body weight, BMI, or FM (daily doses of 300–12.695 IU D3 with or without calcium) [68].

No further arguments supporting a dose–response relationship between adiposity measures and vitamin D3 were found in a meta-analysis. Another meta-analysis including nine RCTs with 1586 obese but otherwise healthy adults was conducted by Manousopoulou et al. and showed that neither body weight nor BMI were influenced by vitamin D supplementation [69].

Epidemiological studies, systematic reviews, and cohort analyses have recently provided us with evidence that reduced vitamin D status is a common finding in obese adults, but even though it seems to be inversely correlated with adiposity measures, the exact mechanism remains unknown. Epidemiological studies represent the sources of the strongest indicators, suggesting an inverse relationship regarding BMI and vitamin D levels. Whether this is a cause or an effect is yet to be determined. This association could potentially be further dampened by multiple confounding factors such as dietary choices, race, ethnicity, lack of sun exposure, and sedentary lifestyles. It is our belief that the mechanisms linking low 25(OH)–vitamin D levels to increased obesity will be better understood in the future thanks to ongoing molecular research on the *CYP2R1* gene, VDR, adipogenesis, and adipolysis [70].

## 7. Conclusions

Vitamin D deficiency is frequently encountered in endocrine disorders, and supplementation restores normal values. In Hashimoto’s disease, low production of vitamin D appears to be related to a higher titer of anti-TPO antibodies and to thyroid volume, and supplementation was associated with a reduction of antibodies in some studies. In other studies, supplementation appeared to reduce TSH levels. In Grave’s disease, the thyroid volume and the degree of exophthalmos revealed a statistically significant correlation with vitamin D levels. In diabetes mellitus type 2 patients, supplementation led to some improvement in the HOMA-IR index and HbA1c. The short-term supplementation treatment of vitamin D in T2DM appeared to have a major impact on HbA1c values, with the receptor response reducing insulin resistance and insulin synthesis in T2DM patients, suggesting that vitamin D can be included as additional therapeutic agent in diabetes for patients with obesity. Another meta-analysis illustrated a reduction in the T2DM prevalence and an increase in the reversion rate of high glycemic values considered prediabetes to normal glycemic values in prediabetic subjects receiving vitamin D supplementation; furthermore, other sources from the literature did not report significant beneficial findings. In obese patients, data from the literature suggest that none of the adiposity measures were decreased by supplementation. Given all this proof of the beneficial role of vitamin D supplementation, our opinion is that vitamin D serum concentration determination and the correction of deficiency are worth implementing in addition to conditional therapies.

## Figures and Tables

**Figure 1 medicina-58-00194-f001:**
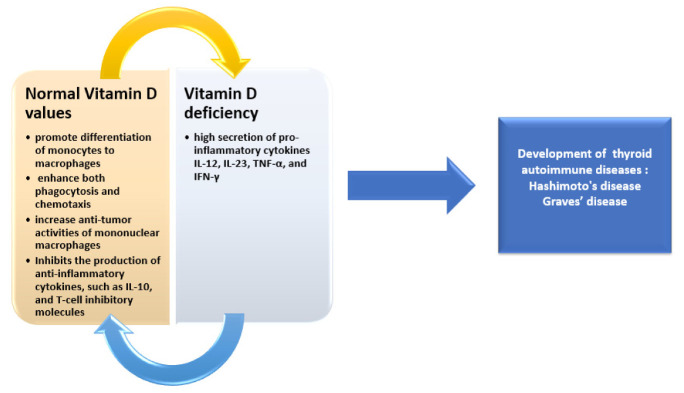
Implications of vitamin D in autoimmune thyroid disorders.

**Figure 2 medicina-58-00194-f002:**
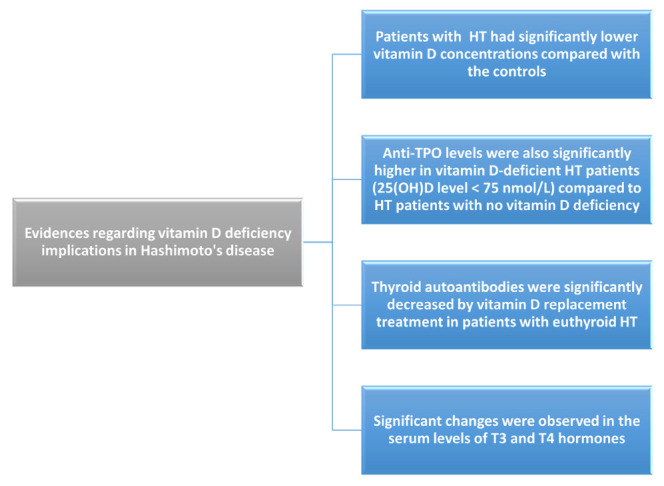
Evidence regarding vitamin D deficiency implications in Hashimoto’s disease.

**Figure 3 medicina-58-00194-f003:**
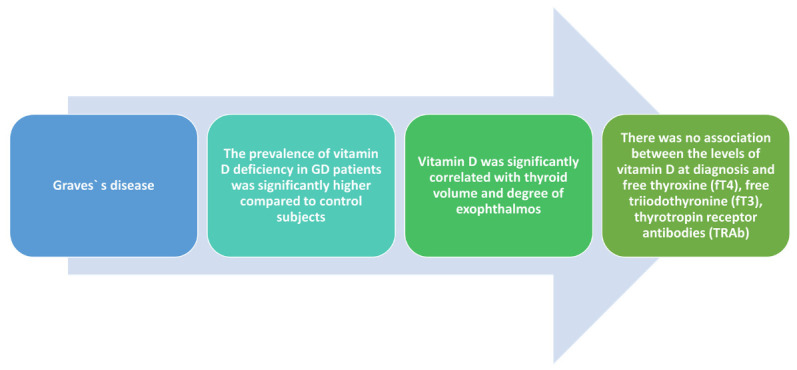
Evidence regarding vitamin D deficiency implications in Grave’s disease.

**Table 1 medicina-58-00194-t001:** Process of bibliographical source selection.

Identification of Studies via Databases and Registers for the Realization of the Review		
Identification	Key words with “AND” operator	“autoimmune thyroid disorders” AND “vitamin D”, “Hashimoto’s diseases” AND “vitamin D”, “Graves’ disease” AND “vitamin D”, “type 2 diabetes mellitus” AND “vitamin D”, “obesity” AND “vitamin D”, etc.
	Consulted databases	Web of Science, PubMed
	Criteria for inclusion	Review or original article, relevant for the topic and thematic
	Criteria for exclusion	Abstract paper, articles where full text was not available, not relevant to the topic or thematic
Records identified	185 records	
Articles after duplicate removal (duplicates = 21)	164 records	
Full-text analysis	Appliance of inclusion and exclusion criteria (eliminated 49 articles)	121 records
Final bibliographical sources	Verification and agreement on article relevance and quality	72 records

## Data Availability

Not appliabile.

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
