# Peer review of "Vitamin D Implications and Effect of Supplementation in Endocrine Disorders: Autoimmune Thyroid Disorders (Hashimoto’s Disease and Grave’s Disease), Diabetes Mellitus and Obesity"

_medicina, 2022, doi:10.3390/medicina58020194_

Round 1

Reviewer 1 Report

In introduction it should be added that similar to obesity , patients with NAFLD tend to have vitamin D deficiency an this is associated with the risk of various infections ( https://pubmed.ncbi.nlm.nih.gov/33977096/)

Methodology- how the authors selected articles- detailed search criteria should be explained

Otherwise I don't have any major comments and I would like to congratulate the authors on this excellent review article 

Author Response

In introduction it should be added that similar to obesity, patients with NAFLD tend to have vitamin D deficiency an this is associated with the risk of various infections ( https://pubmed.ncbi.nlm.nih.gov/33977096/)

Thank you very much for your suggestion, we have added the association and reffered to the above mentioned article in the bibliography.

Methodology- how the authors selected articles- detailed search criteria should be explained

Thank you for the observation, we have added a new chapter called material and methods where we explained the selection process.

Otherwise I don't have any major comments and I would like to congratulate the authors on this excellent review article 

Reviewer 2 Report

Dear author 

This paper explained about vitamin D and metabolic disorder. It is good idea, but the authors do not discuss about other thyroid diseases. Also, what is the reason for choosing diabetes and obesity?

Please add reference for the line of 288- 293

Author Response

Dear author 

This paper explained about vitamin D and metabolic disorder. It is good idea, but the authors do not discuss about other thyroid diseases. Also, what is the reason for choosing diabetes and obesity?

Thank you for this observation, we detailed it here and in the article. We aimed to discuss about autoimmune thyroid pathology, diabetes mellitus and obesity because we work in a tertiary care hospitals, Emergency Clinical County Hospital of Oradea and Gavril Curteanu City Hospital Oradea, Romania where we encounter both in the diabetes clinic, internal medicine clinic, endocrinology clinic most often these above mentioned endocrinological diseases. It is a choice based on our clinical experience. The diseases were chosen according to the observed frequency of their apparition in the hospital that we work in. We explained our choice in the material and methods chapter.

Please add reference for the line of 288- 293

Thanks for the suggestion, we corrected the mistake.